# Current Advances in Corneal Stromal Stem Cell Biology and Therapeutic Applications

**DOI:** 10.3390/cells13020163

**Published:** 2024-01-16

**Authors:** Thomas Volatier, Claus Cursiefen, Maria Notara

**Affiliations:** 1Department of Ophthalmology, Faculty of Medicine, University Hospital Cologne, University of Cologne, 50937 Cologne, Germany; 2Cologne Excellence Cluster for Cellular Stress Responses Aging-Associated Diseases (CECAD), University of Cologne, Joseph-Stelzmann-Strasse 26, 50931 Cologne, Germany; 3Center for Molecular Medicine Cologne (CMMC), Faculty of Medicine, University Hospital Cologne, University of Cologne, 50931 Cologne, Germany

**Keywords:** cornea, stem cells, UV-light, limbal stem cells, regenerative ophthalmology

## Abstract

Corneal stromal stem cells (CSSCs) are of particular interest in regenerative ophthalmology, offering a new therapeutic target for corneal injuries and diseases. This review provides a comprehensive examination of CSSCs, exploring their anatomy, functions, and role in maintaining corneal integrity. Molecular markers, wound healing mechanisms, and potential therapeutic applications are discussed. Global corneal blindness, especially in more resource-limited regions, underscores the need for innovative solutions. Challenges posed by corneal defects, emphasizing the urgent need for advanced therapeutic interventions, are discussed. The review places a spotlight on exosome therapy as a potential therapy. CSSC-derived exosomes exhibit significant potential for modulating inflammation, promoting tissue repair, and addressing corneal transparency. Additionally, the rejuvenation potential of CSSCs through epigenetic reprogramming adds to the evolving regenerative landscape. The imperative for clinical trials and human studies to seamlessly integrate these strategies into practice is emphasized. This points towards a future where CSSC-based therapies, particularly leveraging exosomes, play a central role in diversifying ophthalmic regenerative medicine.

## 1. Introduction

Impairment of corneal transparency and compromised refractive function are prominent causes of blindness. It is estimated that between 4.9 and 5.5 million people worldwide are bilaterally blind or have bilateral visual impairment, both resulting from corneal opacification [1,2,3,4]. It is estimated that an additional 6.2 million people are unilaterally blind. According to the WHO, 1.9 million instances of this opacification of the cornea are due to trachoma, an infection caused by the bacteria Chlamydia trachomatis [5,6]. The remaining cases of corneal opacification are due to a great number of other factors, such as onchocerciasis, vitamin A deficiency, tumors, and wounds [7,8]. An estimated 55.3% of the world’s population does not have access to corneal transplant material [9,10,11,12]. The incidence of corneal blindness is particularly pronounced in countries within Asia, Africa, and the Middle East, where over 1 million new cases arise each year [13,14,15,16]. The impact of prolonged exposure to ultraviolet (UV) radiation, coupled with the prevalence of untreated ocular diseases, contributes to the greater incidence of corneal blindness [15,17,18]. Access to corneal transplantation remains a formidable challenge, affecting more than half of the world’s population due to deficient infrastructure for tissue donation, harvesting, testing, and eye banking [11].

The use of stem cells and stem cell stimulation to restore cornea clarity presents a potential strategy to address the shortage of donor corneas for transplantation by making transplantations a lower priority. In the case of limbal epithelial cells, stem cell transplantation is already a clinical routine [19,20,21,22]. Utilizing the regenerative potential of a patient’s own stem cells eliminates the dependency on external donor sources, mitigating the challenges associated with donor cornea scarcity. Autologous stem cell therapies involve isolating and cultivating a patient’s own stem cells, typically from the limbus or other corneal tissues, and then reintroducing them to the damaged cornea. Additionally, stem cell stimulation techniques, such as the application of growth factors or gene therapies, can activate endogenous corneal stem cells, enhancing their regenerative capabilities [23]. This approach was first demonstrated in 2008, when pancreatic exocrine cells were reprogrammed into insulin-producing beta cells [24]. The same concept applied to the cornea has been explored to suppress the undesirable epithelial–mesenchymal transition of the corneal epithelium in vitro [25,26]. By fostering the natural repair mechanisms within the patient’s own cornea, these approaches offer a potential, sustainable, and patient-specific solution to the shortage of transplantable corneas. This innovative approach not only addresses the limitations of donor availability but also holds the potential to revolutionize corneal transplantation, making it more accessible and feasible for patients worldwide.

The research areas discussed here, including corneal stromal stem cell biology, corneal defects, exosome therapy, and the use of autologous stem cells, hold potential for advancing the field of ophthalmology and corneal care. Understanding corneal stromal stem cells and their molecular markers can lead to more targeted therapies to address corneal diseases and injuries, improving visual outcomes for patients. Research into corneal stromal cell defects and regenerative approaches can lead to innovative treatments that enhance the regenerative potential of the cornea and somewhat mitigate shortages of tissue. Additionally, exosome therapy offers a new, non-invasive means of modulating corneal cell behavior, potentially revolutionizing treatment strategies [27]. Exosome-based approaches have already proven popular with various stem cell-related diseases and conditions, from myocardial infarctions to Alzheimer’s [28,29,30,31,32]. The use of autologous stem cells and stem cell stimulation techniques has the potential to alleviate donor cornea scarcity but can also improve transplant procedures by targeting relevant stem cell behavior, such as immunomodulation, differentiation, or anti-vascularization. These research efforts show promise to improve the field of corneal care, exploring novel solutions and improved outcomes for individuals suffering from corneal disorders and injuries.

## 2. Corneal Anatomy

The cornea, a transparent, avascular tissue forming the anterior part of the eye, protects the contents of the eye and provides the main ocular refractive power. Comprising five distinct layers, which are illustrated in Figure 1, the cornea exhibits a meticulously organized structure, with each layer contributing to its unique optical and biomechanical properties [33,34].

The corneal epithelium, the outermost layer of the cornea, is a highly specialized and transparent tissue made of layers of tightly packed epithelial cells. This layer is the first protective barrier against environmental factors, including pathogens and foreign particles. The corneal epithelium is also the first layer to refract light, contributing to visual acuity [34,35]. This stratified epithelium undergoes continuous renewal, with limbal epithelial stem cells dividing asymmetrically and migrating along the basal layer to then divide further and migrate toward the surface. These fully differentiated individuals eventually slough off [36,37]. Nerve endings from the sub-basal plexus innervate the corneal epithelium, providing sensory input and contributing to the eye’s sensitivity. The integrity and health of the corneal epithelium are crucial for maintaining proper vision, and any disruptions or defects in this outermost layer can lead to visual disturbances and a compromised ocular surface [38,39,40]. Due to several redundant mechanisms, the corneal epithelium also significantly contributes to physiological corneal avascularity [41,42].

Beneath the epithelium lies Bowman’s layer, an acellular, collagenous sheet providing structural support to the cornea. Though relatively thin, it contributes to the cornea’s tensile strength, provides a basement layer to the epithelium, and acts as a barrier against epithelial ingrowth [43]. This layer also houses the sub-basal plexus, which sends nerve fiber branches orthogonally into the anterior epithelium [44,45].

Constituting approximately 80% of the corneal thickness, the stroma is a collagen-rich matrix interspersed with keratocytes. The stroma is an intricate, hydrated extracellular matrix primarily comprised of interwoven fibrous proteins, predominantly collagen types I and V [46,47]. Within the collagen fibers are glycosaminoglycans, such as chondroitin, dermatan, keratan, and heparan sulfates, which play essential roles in regulating hydration, binding cations, and preserving structural integrity [34]. 

Collagen fibrils within the human cornea are organized, forming bundles known as lamellae that intersect across the cornea’s width [48]. In the anterior region, these lamellae exhibit more pronounced interweaving, contributing significantly to the cornea’s shape and transparency, particularly relevant to its refractive properties. The alignment of collagen fibrils also contributes to corneal transparency, creating a pseudo-crystalline structure that facilitates the transmission of light through the cornea [46,47]. The distribution and organization of glycosaminoglycans in the corneal stroma exhibit a gradient along the anterior–posterior axis, resulting in differential responses to stress and swelling in various parts of the corneal stroma [49,50]. Notably, collagen bundles in the mid and posterior stroma display an orthogonal arrangement, while those in the anterior stroma are characterized by thinner and interwoven patterns [51]. 

Situated beneath the stroma, Descemet’s membrane is a basement membrane that provides additional structural support. This acellular layer plays a role in maintaining corneal shape and integrity. It also serves as a substrate for endothelial cell attachment [52].

The innermost layer, the corneal endothelium, is a monolayer of cells that regulates corneal dehydration. Through active ion transport, the endothelium ensures that the stroma remains dehydrated, preventing corneal swelling. Any swelling of the cornea would result in deformation and loss of visual acuity. Unlike other corneal layers, the endothelium has limited regenerative capacity, making its health vital for overall corneal function. Dysfunction of the corneal endothelium is the most common cause of corneal transplantation in industrialized countries [53].

## 3. Identity of Corneal Stromal Stem Cells

### 3.1. Origins and Potency

The mammalian eye undergoes construction in developmental stages originating from the neural crest, which gives rise to three distinct sources of precursor eye cells: the neural ectoderm, surface ectoderm, and periocular mesenchyme [54,55]. Specifically, the periocular mesenchyme contributes to the formation of stromal cells responsible for building the stroma, eventually transforming into keratocytes [56]. Within the periocular mesenchyme, these cells play a vital role in the initial construction and remodeling of the extracellular matrix (ECM) into the stroma, characterized by a unique structure and deposited proteins ensuring transparency [57]. As the tissue develops, apoptosis occurs, leaving behind a small population that expresses crucial proteins for corneal stromal cell function, including crystallins [58]. Tracking the differentiation from neural crest cells to quiescent keratocytes involves observing the disappearance of PAX6, a marker of early eye development, and the emergence of CD34, a general marker of mesenchymal stem cells [59]. The expression of CD34 diminishes as quiescent keratocytes become activated, as notably observed in vitro, where CD34 expression nearly vanishes upon keratocyte activation [60]. 

### 3.2. Molecular Markers

Stromal stem cells in the cornea are characterized by their unique molecular markers and location within the tissue. Corneal keratocytes, residing within the corneal stroma, exhibit distinctive molecular markers that are integral to their identification and functional characterization. Among these markers, Pax6 and ABCG2 serve as key indicators of keratocyte progenitor cells, highlighting their quiescent and stem cell-like properties, respectively. Other general mesenchymal stem cell markers, such as MSC markers such as CD73, CD90, CD105, and CD140b/PDGFRβ, are also expressed [61]. ABCB5, a member of the ATP-binding cassette transporter family, is another noteworthy molecular marker associated with corneal keratocytes, emphasizing their stem cell characteristics [62,63]. Pax6, ABCG2, and ABCB5 are also present in corneal epithelial cells, particularly the corneal epithelial stem cells [59,63]. It is possible to distinguish between stromal and epithelial stem cells as the two are separated by Bowman’s layer and the stromal cells express mesenchymal markers that are not found in epithelial cells [61].

These molecular markers play a pivotal role in distinguishing keratocytes from other corneal cell types, providing valuable insights into their regenerative potential. Understanding the nuanced molecular profile of corneal keratocytes is essential for advancing research on corneal biology, tissue engineering, and therapeutic interventions aimed at enhancing corneal health and regeneration. Previous exploration into MSC conversion to corneal stromal cells achieved corneal stromal cell marker expression [64]. This suggests that if CSSC is not available, MSC can be used to some extent in formulating therapeutic cell sources. 

Corneal stromal stem cells are distinct cell populations with unique characteristics. According to the in vitro behavior of CSSCs, they can be characterized as MSCs according to the guidelines set by the International Society of Cellular Therapy [65]. While both CSSC and MSC have the potential to differentiate into various cell types, there are notable differences between them. They each exhibit specific lineage commitments. Corneal stromal stem cells have the primary capacity to differentiate into corneal keratocytes, which are responsible for the maintenance of the corneal stroma. In contrast, mesenchymal stem cells are a more broadly distributed population found in various tissues, such as bone marrow and adipose tissue. Both of these cells possess in vitro trilineage differentiation potential, meaning they can differentiate into cells from three different lineages: osteogenic, chondrogenic, and adipogenic lineages. This broader differentiation capacity makes MSCs versatile for regenerative applications in various tissues and systems, whereas corneal stromal stem cells are more specialized in their role, specifically contributing to corneal health and transparency. It should be noted that the term MSC covers a very diverse range of isolation origins, each with their own regenerative potency. Corneal research has previously investigated the usage of MSC from several sources, such as the amniotic membrane [66], adipose tissue [67], and bone marrow [68]. 

### 3.3. Location 

Stromal stem cells are primarily situated in the anterior stroma, close to the boundary with the corneal epithelium [69,70,71]. Their strategic positioning allows them to respond promptly to injuries and insults to the cornea. Within the limbal stroma, a niche enriched with extracellular matrix components and signaling molecules, there is a greater number of corneal stromal stem cells that maintain a quiescent state and regenerative potential. These cells serve as a reservoir of progenitor cells that can differentiate into functional keratocytes, the specialized cells responsible for preserving the transparent corneal stroma. Their strategic positioning at the limbus not only allows for rapid response to corneal injuries or insults but also plays a role in preserving the cornea’s vascular privilege. The stromal cells maintain tissue homeostasis through macrophage modulation [72]. Increased macrophage activity, particularly of Cd11c+ macrophages, is one of the precursors to lymphatic vessel appearance in the stroma [73,74]. The prevention of angiogenesis is more direct, with corneal stromal stem cells directly inhibiting vascular endothelial cell sprout formation with anti-angiogenic factors such as sFLT-1 and PEDF [75]. The cornea’s avascular nature is vital for maintaining its transparency, and limbal stromal stem cells may in fact contribute to this avascular privilege by orchestrating the prevention of (lymph)angiogenesis, ensuring that new blood vessel formation is physiologically prevented within the cornea. Understanding the precise location and function of these cells is crucial for developing therapeutic strategies that harness their natural reparative abilities to address corneal issues while preserving the cornea’s optical and physiological integrity. While endogenous MSC are not present in the cornea, they are recruited from the blood to the injured cornea through stem cell chemoattractants substance P and SDF-1 [76]. The recruitment of exogenous MSC from the blood was observed following a limbal injury [76]. When the limbus was intact and only the central cornea was injured, sub-conjunctival injections of MSC were found to have a more potent effect [77]. 

### 3.4. Homeostasis and Tissue Integrity

Even in the absence of injury, stromal stem cells help maintain corneal homeostasis by monitoring the health of keratocytes and participating in the renewal of the stromal matrix. This continual surveillance ensures the cornea’s structural and functional integrity. Stromal stem cells play a key role in ensuring corneal transparency and structural integrity by participating in a bidirectional crosstalk with the corneal epithelium, which helps regulate the renewal of epithelial and stromal cells. This ongoing maintenance by stromal stem cells is essential for preventing the accumulation of structural deficiencies that might otherwise affect the cornea’s transparency and optical quality, thus safeguarding its overall function and visual acuity. 

### 3.5. Regeneration of Keratocytes 

When the cornea is injured or experiences cellular turnover, stromal stem cells are activated to proliferate and differentiate into keratocytes. Keratocytes are the specialized cells responsible for maintaining the extracellular matrix of the corneal stroma, which is primarily composed of collagen fibrils. This regenerative process is essential for restoring the structural integrity and transparency of the cornea.

### 3.6. Maintenance of Corneal Clarity 

Stromal stem cells continuously contribute to corneal clarity by replenishing damaged or aged keratocytes. This ongoing turnover ensures that the cornea remains transparent and free from opacities, which could compromise visual acuity. Corneal keratocytes are responsible for producing and organizing the extracellular matrix, a critical component of corneal transparency. It should be noted that there is a decrease in epithelial stem cell proliferative capacity as patients age. Telomere length and marker expression remain unchanged, at least in corneal epithelial stem cells [78]. This remains to be investigated in corneal stromal stem cells.

The cornea lacks the typical lymphatic vasculature observed in other tissues, contributing to the concept of ocular immune privilege. However, recent evidence, including the identification of lymphatic-like vessels such as Schlemm’s canal, challenges this notion [79]. Under normal conditions, the cornea remains free of blood and lymphatic vessels [80], but inflammatory stimuli, like transplant procedures, can induce lymphatic vessel invasion [81]. The historical notion of an alymphatic cornea was influenced by the absence of specific markers for lymphatic endothelium and their invisibility in conventional examinations [82]. Recent advancements, such as the discovery of molecular markers like LYVE-1, PROX-1, podoplanin, and FLT4, have significantly advanced lymphangiogenesis research [83,84,85,86]. Recent findings reveal the unexpected involvement of lymphatic vessels within the corneal stroma in various ocular pathologies, from dry eye disease to immune responses [42,87,88,89].

In cases of corneal injuries, infections, or degenerative conditions, stromal stem cells play a critical role in the corneal healing process. They respond to signals from damaged tissue and migrate to the site of injury, where they contribute to tissue repair and regeneration.

## 4. Interactions within the Stromal Stem Cell Niche

The most obvious function of the epithelium with regards to the stroma is as a barrier. The epithelium is the first line any exterior threat, be it viral, bacterial, or physical, must traverse before entering the stroma.

Corneal stromal stem cells interact with each other within the stromal matrix (see Figure 2A), arranging a dynamic balance that contributes to tissue homeostasis and repair [90,91]. Cell-to-cell communication among stromal stem cells involves various signaling pathways and molecular cues, influencing their quiescence, proliferation, and differentiation [92]. Additionally, these stromal stem cells interact closely with corneal epithelial cells [93]. Paracrine signaling between stromal and epithelial cells plays a crucial role in maintaining the health and transparency of the cornea [94,95]. Stromal cells provide essential growth factors and extracellular matrix components that support the survival and functionality of the corneal epithelium. This bidirectional crosstalk, sometimes involving immune cells, is essential for the overall integrity and function of the cornea [96].

The interaction between genes and stromal–epithelial crosstalk is crucial for maintaining corneal homeostasis. Key genes, including Krüppel-like factor-4 (Klf4) [97], Pax6, and the Inhibitor of differentiation (Id), are pivotal in regulating the dynamic communication between the corneal stroma and epithelium.

Klf4, a highly expressed transcription factor in the cornea, plays a central role in the stromal–epithelial crosstalk [98]. It regulates epithelial integrity and permeability, ensuring effective communication between corneal layers. Experiments in transgenic mice with reduced Klf4 expression resulted in increased epithelial cell layers and a disrupted barrier function, highlighting Klf4’s importance in maintaining the equilibrium between the stroma and epithelium [99]. This change is linked to the process of epithelial-to-mesenchymal transition [100]. The loss of Klf4 expression can engender worsening of stromal wounds and atypical transformation of keratocytes into myofibroblasts [101,102,103].

The Pax6 gene, known for its involvement in eye development, also influences stromal–epithelial crosstalk [104,105]. Changes in Pax6 levels in transgenic mice led to alterations in the morphology of epithelial, stromal, and endothelial cells, affecting cell adhesion and hydration [106]. This demonstrates how Pax6 variations disrupt the interplay between corneal layers. In humans, mutations in the Pax6 gene are linked to aniridia and aniridia-related keratopathies [107,108,109].

In addition to Klf4 and Pax6, the Inhibitor of differentiation (Id) genes contribute to stromal–epithelial crosstalk [110]. Experiments in vitro found that Id genes regulate cell proliferation and differentiation and are differentially expressed in corneal fibroblasts compared to myofibroblasts [111]. Experiments in vivo found that signaling molecules TGF-β1 and BMP-7 modulate Id gene expression, emphasizing their role in mediating interactions between different cell types within the cornea [112].

Genes like Klf4, Pax6, and Id genes are integral to the stromal–epithelial crosstalk in the cornea. Their roles in maintaining corneal integrity and homeostasis are closely connected to the communication between stromal and epithelial cells. Understanding this crosstalk is important for addressing corneal issues and developing potential therapies to restore corneal health and function, making it a central focus in corneal research and treatment.

Following a wound that exclusively damages the corneal epithelium, interactions between corneal stromal stem cells and the remaining epithelial cells intensify [113]. Stromal stem cells are activated in response to signals released by the injured epithelium, prompting them to proliferate and migrate toward the wound site [114,115]. These cells contribute to the regeneration of the damaged epithelium, ensuring the restoration of a functional and protective outer layer. The coordinated response between stromal stem cells and epithelial cells becomes crucial in maintaining the cornea’s barrier function and preventing further complications.

In the event of a more extensive injury that damages both the corneal epithelium and the underlying stroma (See Figure 2B), the interactions between corneal stromal stem cells and epithelial cells become even more complex [116,117,118]. Stromal stem cells play a multifaceted role by not only participating in the regeneration of the epithelium but also contributing to stromal repair [119]. Their ability to differentiate into keratocytes and modulate the extracellular matrix becomes paramount in restoring the structural integrity of the cornea. Simultaneously, epithelial cells engage in reparative processes, and the interplay between these cell populations becomes pivotal in orchestrating a comprehensive tissue-repair response. Successful interactions between corneal stromal stem cells and epithelial cells following such injuries are essential for restoring the cornea’s functionality and preserving visual acuity.

## 5. Stromal Stem Cell Defects

Corneal stromal stem cell defects constitute a significant area of concern, impacting the regenerative capacity and maintenance of corneal health. These defects may arise from various factors, including genetic mutations or environmental influences, leading to a compromised ability of stromal stem cells to proliferate and differentiate effectively. The precise role stromal stem cell defects play in various corneal stromal diseases (See Figure 2C)—be it hereditary or acquired—remains to be studied [120]. When these stem cells fail to execute their regenerative functions appropriately, the corneal stroma may undergo structural changes, affecting transparency and biomechanical integrity. Such defects can contribute to the development of corneal diseases like stromal dystrophies or may impede the natural healing response following injuries or surgeries, potentially leading to impaired vision by loss of stromal transparency.

Understanding the molecular basis of corneal stromal stem cell defects is paramount for developing targeted therapeutic interventions. It involves deciphering the intricate signaling pathways and genetic factors that regulate the behavior of these stem cells. Defects in signaling molecules or disruptions in the microenvironment surrounding stromal stem cells can hinder their normal function. Emerging technologies, such as gene editing and stem cell therapies, offer promising avenues for addressing these defects, with the potential to enhance the regenerative capabilities of corneal stromal stem cells. Comprehensive research into the underlying causes and mechanisms of corneal stromal stem cell defects holds the key to advancing regenerative medicine approaches for treating corneal disorders and improving overall visual outcomes.

Stromal stem cell defects in the cornea can instigate a cascade of events leading to opacification and vascularization, profoundly affecting visual clarity and corneal health. The corneal stromal stem cell is unique among the other stem cells in the cornea as it must self-renew, direct anti-fibrotic wound healing, and immunomodulate [121,122]. The corneal stromal stem cells also differ from their descendent cells, the corneal fibroblasts. Both are capable of producing collagen-fibrillar ECM. However, it has been observed that the corneal stromal stem cells produce the classical orthogonally-oriented collagen fibrils with components unique to the cornea, such as lumican and keratocan, while corneal fibroblasts tend to produce ECM more similar to scar tissue [123]. In human patients, corneal scars manifest after infection, thermal damage, or chemical injuries. In these scenarios, the accompanying inflammatory response inflicts damage on the corneal stromal layer. When stromal keratocytes undergo injury, inflammatory cytokines and fibrogenic growth factors like TGFb1, TGFb2, and connective tissue growth factor (CTGF), along with downstream signaling such as SMAD and SMAD-independent pathways, activate the CSSC, initiating a wound healing response [124]. Prolonged release of these profibrotic factors prompts excessive differentiation of CSSC into fibroblasts and myofibroblasts, which actively release extracellular matrix, ultimately resulting in the formation of stromal scars [125,126,127]. Excess differentiation and depletion of the CSSC population may occur in particularly severe injuries or in extended microbial infections.

The depletion of CSSC also leads to immunological issues. Neutrophils are part of the rapid response against infection or harmful agents [128]. The infiltration of neutrophils in the cornea is repressed by CSSC secretion of TSG-6, and in the absence of this suppression, scar tissue forms [129]. This is believed to occur due to excessive neutrophil elastase promoting CSSC differentiation [130].

When stromal stem cells fail to properly regulate the production and organization of extracellular matrix components, the corneal stroma may undergo pathological changes, resulting in increased light scattering and opacity. Additionally, the compromised regenerative potential of stromal stem cells may trigger an abnormal vascular response, a condition known as corneal neovascularization. In this process, new blood vessels infiltrate the cornea, disrupting its normally avascular nature. The presence of these vessels not only compromises the cornea’s transparency but also introduces inflammatory and immune responses, exacerbating the overall pathology [131]. Stromal stem cell defects that culminate in opacification and vascularization underscore the critical role these cells play in maintaining the delicate balance of corneal homeostasis and emphasize the need for targeted interventions to preserve the optical and physiological properties of the cornea.

## 6. Stromal Stem Cell Therapy—Potential for Exosome Therapy

Exosome therapy has emerged as a promising avenue for harnessing the therapeutic potential of corneal stromal stem cells. Both by replicating corneal stromal stem cell secretions to target epithelial cells or by affecting CSSC directly [132,133]. Exosomes are a type of extracellular vesicles released by various cell types, including stromal stem cells, that carry a cargo of bioactive molecules, such as proteins, lipids, and nucleic acids [134]. When applied to corneal stromal stem cells, exosomes can act as messengers, facilitating intercellular communication and inducing certain cellular behaviors [135]. Studies have shown that exosomes derived from stromal stem cells can enhance the regenerative properties of recipient cells by promoting cell proliferation, modulating inflammation, and regulating extracellular matrix remodeling (See Figure 2D) [133]. The exosomes secreted by functional corneal stromal stem cells contain specific miRNA shown to be directly linked to a reduction in corneal scarring [136]. Cell-free Exosome therapeutic solutions are in pre-clinical development and face some challenges with regards to potency, scalability, and repeatability [137,138]. These aspects will have to be addressed before clinical trials for corneal applications can begin, although these particular criteria have already been met for other exosomal applications currently in clinical trials [139]. Encouragingly, there are already some clinical trials for exosome therapy on the cornea: eyedrops with MSC-derived exosomes on dry eye in patients with GVHD by Zhou et al. (www.clinicaltrials.gov (accessed on 15 January 2024), NCT04213248) and exosome eye drops as a treatment for dry eye disease by Gao et al. (www.clinicaltrials.gov (accessed on 15 January 2024), NCT05738629). In the context of corneal stromal stem cell defects, exosome therapy holds potential for promoting tissue repair, improving transparency, and mitigating aberrant vascularization. The minimally invasive nature of exosome administration also presents a favorable aspect for therapeutic applications in corneal disorders, providing a novel strategy for optimizing the regenerative capabilities of corneal stromal stem cells.

Furthermore, exosome therapy for corneal stromal stem cells aligns with the paradigm of regenerative medicine, offering a more targeted and precise approach to address specific cellular deficiencies. By delivering bioactive factors directly to the affected cells, exosomes can modulate the molecular and cellular environment, fostering a conducive milieu for enhanced tissue regeneration. This therapeutic strategy holds particular promise in treating conditions where corneal stromal stem cell defects lead to opacification and vascularization, as exosomes can potentially mediate anti-angiogenic effects and promote the restoration of corneal transparency. As research in exosome therapy continues to advance, it opens up exciting possibilities for developing innovative treatments that harness the innate regenerative capabilities of corneal stromal stem cells for improved clinical outcomes.

Mesenchymal stromal cell exosomes have shown some potential with regards to immunomodulatory treatments. Initial work conducted focusing on conditions like Type 1 diabetes, graft-versus-host disease, and organ transplantation hints at the potential of MSC exosomes as a valuable immunomodulatory therapy. However, their inconsistent efficacy and concerns regarding their immunogenicity pose substantial challenges to their clinical application. Interestingly, emerging research indicates that exosomes, the paracrine effectors, derived from MSC have a similar therapeutic benefit to the MSCs themselves [140,141]. Current work already suggests that MSC exosomes can improve the recovery outcomes of corneal insults [142,143]. 

## 7. Stromal Stem Cell Rejuvenation

Corneal stromal stem cell rejuvenation is a novel approach in regenerative medicine. With millions of people worldwide awaiting corneal transplantation and the significant challenges associated with donor cornea scarcity, strategies focusing on the rejuvenation of corneal stromal stem cells offer a potential future paradigm shift: Regeneration instead of Transplantation. The process would involve stimulating the proliferation and differentiation of existing stromal stem cells within the cornea, potentially replenishing the regenerative pool and enhancing the tissue’s inherent capacity for repair. Various techniques, including growth factors, gene therapy, and tissue engineering, are being explored to invigorate corneal stromal stem cells, promoting their functionality and augmenting the overall regenerative potential of the cornea. Epigenetic reprogramming can rejuvenate the stem cell population. The activation of the Yamanaka factors (OCT4, SOX2, KLF4, and MYC) [144] within differentiated cells leads to the induction of pluripotency, giving rise to induced pluripotent stem cells [145]. These stem cells could be used to replenish depleted stromal stem cell niches in patients. A less direct method involves rescuing the stem cell environment, the niche. The combination of paracrine signaling, ECM biophysical properties, soluble factors, and nutrients all combine to provide the stem cells with the necessary cues to fulfill their functions. The modulation of the corneal stem niche has already been approached for limbal epithelial stem cells, the close neighbors of CSSC. These methods include bio-scaffolds, bio-active soluble factor cocktails, and cell-based strategies [146,147,148]. Additional clinical trials and human studies are necessary to integrate these innovative strategies into standard clinical practice. 

Beyond these methods to rejuvenate stem cell populations, there are other, even less direct approaches. These involve complex, systemic interactions such as diet and circadian rhythm. Such approaches are studied, particularly with regards to more general age-related stem cell depletions [149]. These concepts remain to be fully explored with regards to CSSC.

By rejuvenating corneal stromal stem cells, researchers and clinicians aim to not only alleviate the burden of corneal blindness—especially relevant in lower-income countries-but also circumvent the challenges associated with traditional corneal transplantation. The scarcity of donor corneas, particularly in regions with limited infrastructure for tissue donation and eye banking, poses a substantial barrier to addressing the global demand for corneal grafts. Rejuvenating endogenous stromal stem cells offers a self-sustaining and potentially more scalable solution, reducing dependence on external donor sources. This innovative approach, if successful, could revolutionize corneal care, providing a viable alternative for patients in need of corneal tissue replacement and contributing to a more equitable and accessible solution for the treatment of corneal disorders on a global scale.

## 8. Future Research and Therapy Perspectives

The future of corneal stromal stem cell research holds promise for practical applications in treating corneal disorders. Ongoing studies focus on harnessing the regenerative capabilities of these cells to develop targeted interventions for issues such as corneal transparency loss and refractive function impairment. Understanding the characteristics of stromal stem cells establishes a foundation for practical and effective therapeutic approaches.

Moving forward, research should aim to precisely locate exosomes derived from corneal stromal stem cells in both health and disease. Investigating specific microenvironmental interactions will enhance comprehension and pave the way for more targeted therapeutic applications. Exosomes offer a low-invasion biomarker to detect, screen, and diagnose diseases [150]. The abundance of exosomes in tear fluid makes these vesicles a very accessible molecular diagnostic tool [151].

Efforts to identify better markers continue, seeking to refine the identification and characterization of corneal stromal stem cells. Improved markers will facilitate more accurate isolation and study of these cells, contributing to the development of advanced therapeutic strategies. There are several cell types that exosomes can be isolated from, and many environmental stimuli can be used to control exosome release. The searching in this catalog of exosome donors is first refined by markers for the desirable molecules that will determine the vesicles performance in vivo [152,153]. The exosome source affects the exosome markers, which then guide the exosomes functionality [100]. Each of these steps will undoubtedly be the subject of future studies. With future clinical trials in mind, the reproducibility and consistent performance of exosome treatment have to be a goal. This can be achieved with the appropriate immortalized cell line, one that will have to be chosen with particular care for the needs of the corneal stroma [138]. 

Molecular stimulation of corneal stromal stem cells is a frontier in research, aiming to optimize their regenerative potential. Understanding how molecular cues impact the behavior and function of these cells will be pivotal in enhancing their efficacy in therapeutic applications. This is particularly relevant where these stem cells may get transplanted into donors, especially their exosome secretions once embedded in a donor. 

Additionally, research will explore diseases that damage exosomes derived from corneal stromal stem cells, unraveling the interplay between pathological conditions and the regenerative potential of these extracellular vesicles. This knowledge will inform targeted interventions for specific diseases and expand the scope of corneal stromal stem cell-based therapies.

## Figures and Tables

**Figure 1 cells-13-00163-f001:**
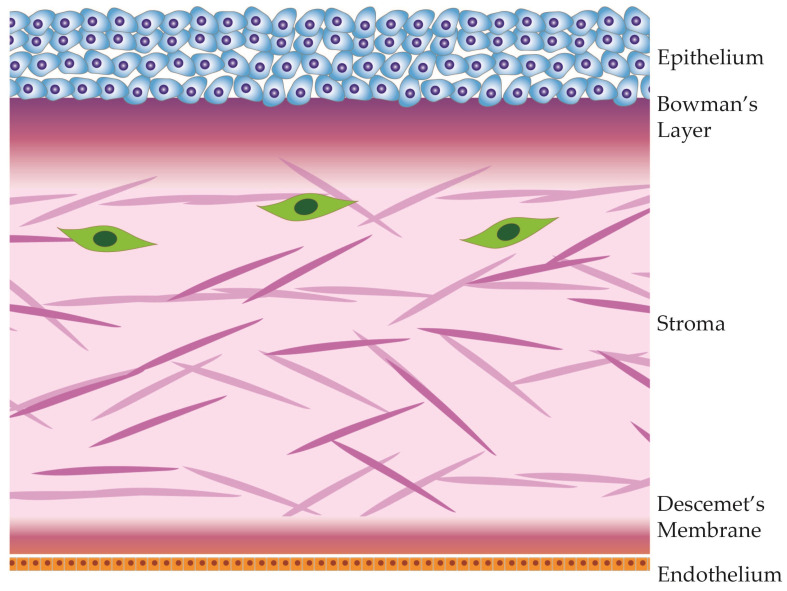
Anatomical diagram of the cornea and its 5 distinct layers.

**Figure 2 cells-13-00163-f002:**
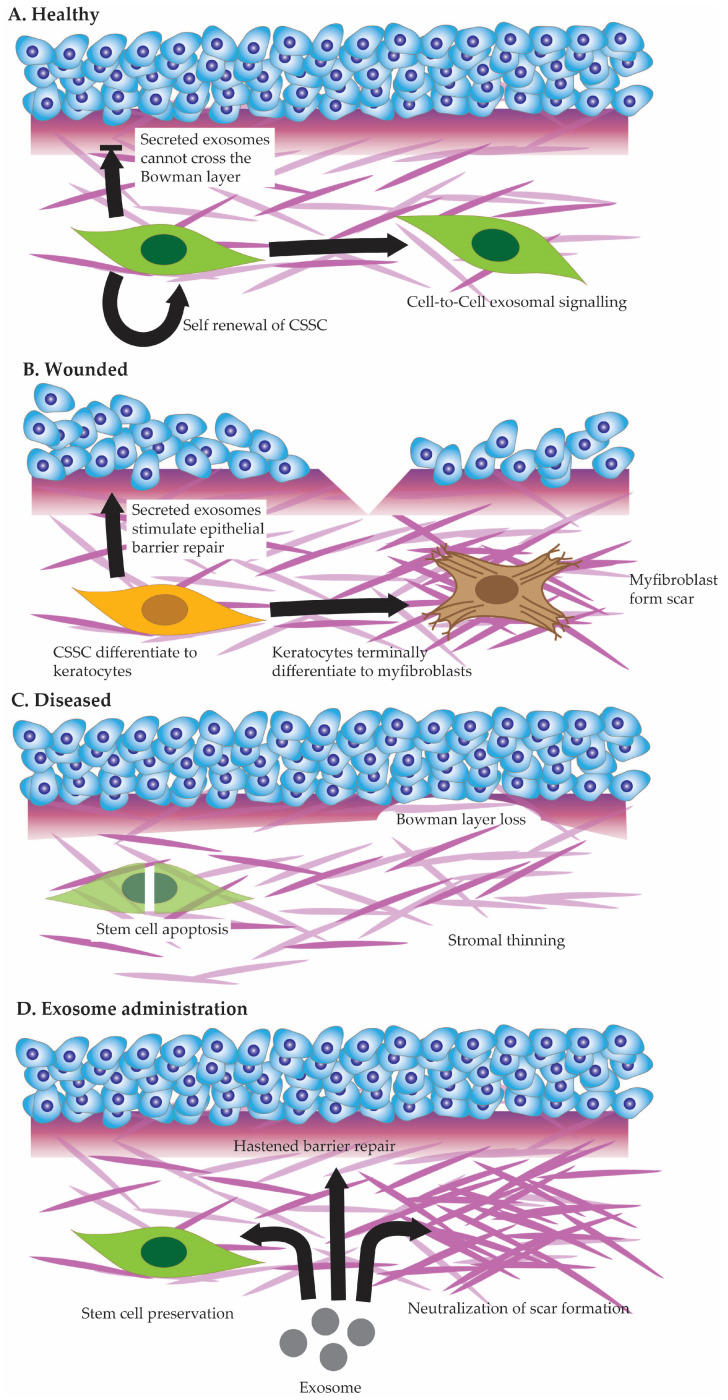
Cell state in healthy cornea (**A**), injured cornea (**B**), and diseased cornea (**C**). Potential future benefits of exosome administration to corneal healing (**D**).

## Data Availability

No new data were created or analyzed in this study. Data sharing is not applicable to this article.

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
