# Peer review of "Current Advances in Corneal Stromal Stem Cell Biology and Therapeutic Applications"

_cells, 2024, doi:10.3390/cells13020163_

Round 1
Reviewer 1 Report
Comments and Suggestions for Authors
This review provides a detailed description of corneal stromal stem cells CSSCs, their anatomy, functions, and role in maintaining corneal integrity, as well as molecular markers, wound healing mechanisms, and potential therapeutic applications. Of particular interest to the authors is exosome therapy. CSSC-derived exosomes could have a significant potential in modulating inflammation, promoting tissue repair, and helping maintain corneal transparency. Additionally, the rejuvenation potential of CSSCs through epigenetic reprogramming adds to the evolving regenerative landscape. The need for clinical trials and human studies to integrate these strategies into practice is emphasized. The review is well written, albeit a little too enthusiastic about exosomes that have not found their way into clinic yet. Certain aspects of stem cell use and limitations of exosome therapy need to be covered in more detail.
This reviewer has the following concerns about the paper.
1. The abstract has a lot of "big words" (compelling avenue, revolutionizing) that may need to be toned down because most of the related strategies remain at the preclinical stage.
2. Lines 168-169. The authors claim that the markers mentioned before can distinguish keratocytes from other corneal cells. However, ABCG2, ABCB5, PAX6 are also expressed by epithelial cells.
3. Lines 178-191. MSCs are not created equal. Different isolates have different regenerative potencies, which has been abundantly described. Some words to this effect would help the readers understand the complexity of using MSCs for a specific tissue.
4. Lines 236-238. The replenishment of aged keratocytes is a problem. Female transplants retain Barr bodies for decades in keratocytes.
5. Line 348. Keratan sulfate does not occur by itself and can be found in cartilage and neural cells. Please remove.
6. Lines 388-389. The non-invasive administration applies to cell-cell communication. It is unclear whether this would directly apply to therapy.
7. Lines 379-403. Please add some references.
8. The review would benefit from the discussion of using stromal stem cells to fight scar formation. Any ongoing clinical trials should be mentioned.
9. Likewise, clinical trials for exosome therapy should be mentioned, as well as significant current drawbacks in the implementation of this promising therapy including lack of standardization, lack of GMP guidelines, difficulties in scaling-up, heterogeneity, potential effect dependence on the source. These considerations need to be addressed in the text and conclusions.
Author Response
Thank you for your constructive comments. We have amended your suggestions to the best of our abilities in order to improve the manuscript as outlined below.
This reviewer has the following concerns about the paper.
- The abstract has a lot of "big words" (compelling avenue, revolutionizing) that may need to be toned down because most of the related strategies remain at the preclinical stage.
The language in the abstract has been toned down to better reflect the current state of research.
Changed “compelling avenue” to “potential” on line 18
Changed “revolutionizing” to “diversifying” on line 23
- Lines 168-169. The authors claim that the markers mentioned before can distinguish keratocytes from other corneal cells. However, ABCG2, ABCB5, PAX6 are also expressed by epithelial cells.
Added “Pax6, ABCG2, and ABCB5 are also present in corneal epithelial cells, particularly the corneal epithelial stem cells [59, 63]. It is possible to distinguish between the stromal and epithelial stem cells as the two are separated by Bowman’s layer and the stromal cells express MSC markers that are not recorded in epithelial [61].” On line 168
- Lines 178-191. MSCs are not created equal. Different isolates have different regenerative potencies, which has been abundantly described. Some words to this effect would help the readers understand the complexity of using MSCs for a specific tissue.
Added “It should be noted that the term MSC covers a very diverse range of isolation origins, each with their own regenerative potency. Corneal research has previously investigated the usage of MSC from several sources such as: amniotic membrane [66], adipose [67], and bone-marrow [68].” At line 195
- Lines 236-238. The replenishment of aged keratocytes is a problem. Female transplants retain Barr bodies for decades in keratocytes.
Added “It should be noted that there is a decrease in epithelial stem cell proliferative capacity as patients age. Telomere length and marker expression remains unchanged, at least in corneal epithelial stem cells [74]. This remains to be investigated in corneal stromal stem cells.” At line 245
- Line 348. Keratan sulfate does not occur by itself and can be found in cartilage and neural cells. Please remove.
Removed mention of keratan sulfate
- Lines 388-389. The non-invasive administration applies to cell-cell communication. It is unclear whether this would directly apply to therapy.
Removed “immense” from “exosome therapy holds immense potential” on line 408
Changed “non-invasive” to “minimally invasive” on line 409
Ongoing clinical trials that use exosomes to target the cornea use eyedrops (See point 8 on the list). Approaches that would target the stroma exclusively may use injections, to better reflect this possibility, “non-invasive” was changed to “minimally invasive”.
- Lines 379-403. Please add some references.
Changed “Exosomes are small extracellular vesicles” to “Exosomes are a type of extracellular vesicles” and added a relevant reference on line 393
Changed “influencing cellular behaviors” to “inducing certain cellular behaviors” and added a relevant reference on line 394
Added a relevant reference on line 397
- The review would benefit from the discussion of using stromal stem cells to fight scar formation. Any ongoing clinical trials should be mentioned.
Added examples of clinical trials targeting the cornea, neither trial has published articles listed. While the clinical trials listed do target the cornea, they do not explicitly target the stromal stem cells.
Added “The exosomes secreted by functional corneal stromal stem cells contain specific miRNA shown to be directly linked to a reduction in corneal scarring [132]. Cell-free Exosome therapeutic solution are in pre-clinical development and face some challenges with regards to potency, scalability, and repeatability [133, 134]. These aspects will have to be addressed before clinical trials for corneal applciations can begin, although these particular criteria have already been met for other exosomal application currently in clinical trials [135]. Encouragingly, there already are some clinical trials for exosome therapy on the cornea: MSCs derived exosomes on dry eye in patients with GVHD by Zhou et al. (www.clinicaltrials.gov, NCT04213248) and exosome eye drops as a treatment for dry eye disease by Gao et al. (www.clinicaltrials.gov, NCT05738629).” On line 397
A survey of clinicaltrials.gov shows that there are not ongoing clinical trials using exosomes to reduce corneal scarring specifically
- Likewise, clinical trials for exosome therapy should be mentioned, as well as significant current drawbacks in the implementation of this promising therapy including lack of standardization, lack of GMP guidelines, difficulties in scaling-up, heterogeneity, potential effect dependence on the source. These considerations need to be addressed in the text and conclusions.
Added “The exosomes secreted by functional corneal stromal stem cells contain specific miRNA shown to be directly linked to a reduction in corneal scarring [132]. Cell-free Exosome therapeutic solution are in pre-clinical development and face some challenges with regards to potency, scalability, and repeatability [133, 134]. These aspects will have to be addressed before clinical trials for corneal applciations can begin, although these particular criteria have already been met for other exosomal application currently in clinical trials [135]. Encouragingly, there already are some clinical trials for exosome therapy on the cornea: MSCs derived exosomes on dry eye in patients with GVHD by Zhou et al. (www.clinicaltrials.gov, NCT04213248) and exosome eye drops as a treatment for dry eye disease by Gao et al. (www.clinicaltrials.gov, NCT05738629).” On line 397
Added “With future clinical trials in mind, the reproducibility and consistent performance of exosome treatment has to be a goal. This can be achieved with the appropriate immortalized cell line, one which will have to be chosen will particular care to the needs of the corneal stroma [134].” On line 496
Reviewer 2 Report
Comments and Suggestions for Authors
1. The paper emphasizes the advantages of emerging exosome therapy, yet does not mention its drawbacks, this kind of discussion is not comprehensive. I think a table can be created to compare it with other current therapies, which can more clearly reflect which problems have been solved by exosome therapy in old methods, and what problems does it have.
2. Moreover, the manuscript discusses a lot about the therapeutic applications of CSSC exosomes, but it seems still lack of the part of the introduction about CSSC isolation and culture techniques. I think this is important, because cell expansion efficiency greatly influence whether this cell types could be widely applied.
3. From Line 201 to 209. The paper mentioned that CSSC may prevent the formation of lymphatic vessels and blood vessels. The relevant researches should be listed and the mechanisms should be briefly explained.
4. Line 383.“Studies have shown that……” Please add the relevant papers on the corresponding effects of stromal stem cell exosomes on recipient cells.
5. In the section "Interactions within the stromal stem cells niche", the authors should first explain the interactions between CSSCs and other cells (Line 290 to 318), and then introduce that among these interactions, what are the key genes and factors (264-289). This is more logical.
Comments on the Quality of English LanguageThe quality of the written language is good.
Author Response
- The paper emphasizes the advantages of emerging exosome therapy, yet does not mention its drawbacks, this kind of discussion is not comprehensive. I think a table can be created to compare it with other current therapies, which can more clearly reflect which problems have been solved by exosome therapy in old methods, and what problems does it have.
As there are no currently available therapies that use exosomes to target the cornea, we have not made direct comparisons to existing treatments. Instead we have listed current clinical trials that use exosomes on cornea. We also listed problems with designing exosome therapies.
Added “Cell-free Exosome therapeutic solution are in pre-clinical development and face some challenges with regards to potency, scalability, and repeatability [137, 138]. These aspects will have to be addressed before clinical trials for corneal applciations can begin, although these particular criteria have already been met for other exosomal application currently in clinical trials [139]. Encouragingly, there already are some clinical trials for exosome therapy on the cornea: MSCs derived exosomes on dry eye in patients with GVHD by Zhou et al. (www.clinicaltrials.gov, NCT04213248) and exosome eye drops as a treatment for dry eye disease by Gao et al. (www.clinicaltrials.gov, NCT05738629).” On line 404. Moreover, the manuscript discusses a lot about the therapeutic applications of CSSC exosomes, but it seems still lack of the part of the introduction about CSSC isolation and culture techniques. I think this is important, because cell expansion efficiency greatly influence whether this cell types could be widely applied.
Added “With future clinical trials in mind, the reproducibility and consistent performance of exosome treatment has to be a goal. This can be achieved with the appropriate immortalized cell line, one which will have to be chosen will particular care to the needs of the corneal stroma [138].” On line 501
- From Line 201 to 209. The paper mentioned that CSSC may prevent the formation of lymphatic vessels and blood vessels. The relevant researches should be listed and the mechanisms should be briefly explained.
Added “. The stromal cells maintain tissue homeostasis through macrophage modulation [72]. Increased macrophage activity, particularly of Cd11c+ macrophages, is one of the precursors of lymphatic vessel appearance in the stroma [73, 74]. The prevention of angiogenesis is more direct, with corneal stromal stem cells directly inhibiting vascular endothelial cell sprout formation with antiangiogenic factors such as sFLT-1 and PEDF [75].” On line 210
- Line 383.“Studies have shown that……” Please add the relevant papers on the corresponding effects of stromal stem cell exosomes on recipient cells.
Changed “Exosomes are small extracellular vesicles” to “Exosomes are a type of extracellular vesicles” and added a relevant reference on line 393
Changed “influencing cellular behaviors” to “inducing certain cellular behaviors” and added a relevant reference on line 394
Added a relevant reference on line 397
- In the section "Interactions within the stromal stem cells niche", the authors should first explain the interactions between CSSCs and other cells (Line 290 to 318), and then introduce that among these interactions, what are the key genes and factors (264-289). This is more logical.
Moved the introductory section from line 315 to line 275, prior to the listing of key genes and factors.
Reviewer 3 Report
Comments and Suggestions for Authors
This is a well compiled review on corneal stromal stem cells and the role of exosomes. There are no major concerns with the review. It would have been more clinically relevant if the authors would add the mode of delivery for therapeutic application of exosomes or CSSC transplantation for corneal diseases. One additional aspect, the authors could add is the usage of various carriers that could be explored for transplantation of cells or cell free exosomes for delivery. These two points could be added in the future applications.
Author Response
Thank you for the positive evaluation of our manuscript and your constructive comments.
To address the points raised, we added examples of clinical trials and exosome delivery methods on line 414 with: “Cell-free Exosome therapeutic solution are in pre-clinical development and face some challenges with regards to potency, scalability, and repeatability [137, 138]. These aspects will have to be addressed before clinical trials for corneal applications can begin, although these particular criteria have already been met for other exosomal application currently in clinical trials [139]. Encouragingly, there already are some clinical trials for exosome therapy on the cornea: eyedrops with MSC-derived exosomes on dry eye in patients with GVHD by Zhou et al. (www.clinicaltrials.gov, NCT04213248) and exosome eye drops as a treatment for dry eye disease by Gao et al. (www.clinicaltrials.gov, NCT05738629).”
Added some more specific design considerations: “With future clinical trials in mind, the reproducibility and consistent performance of exosome treatment has to be a goal. This can be achieved with the appropriate immortalized cell line, one which will have to be chosen will particular care to the needs of the corneal stroma [138].” On line 501
Round 2
Reviewer 1 Report
Comments and Suggestions for Authors
The answers were fully adequate. In the final version, please correct the sentence on line 168 "the stromal cells express MSC markers that are not recorded in epithelial" to "the stromal cells express mesenchymal markers that are not found in epithelial cells"
Reviewer 2 Report
Comments and Suggestions for Authors
In general, the article has been greatly improved after the revision. In my opinion, the safety and efficacy of using immortalized cell lines to produce exosomes still need to be further studied. In theory, corneal stromal stem cells should have a good ability to expand in vitro. Therefore, in the idea of producing exosomes, focusing on optimizing the culture technology of stem cells to maintain their stem cell characteristics and function in vitro could be a better solution, because the secretion and stem cell characteristics of genetically modified immortalized cells may have great changes.